# The Association of Nighttime Fasting Duration and Prostate Cancer Risk: Results from the Multicase-Control (MCC) Study in Spain

**DOI:** 10.3390/nu13082662

**Published:** 2021-07-30

**Authors:** Anna Palomar-Cros, Ana Espinosa, Kurt Straif, Beatriz Pérez-Gómez, Kyriaki Papantoniou, Inés Gómez-Acebo, Ana Molina-Barceló, Rocío Olmedo-Requena, Juan Alguacil, Guillermo Fernández-Tardón, Delphine Casabonne, Nuria Aragonés, Gemma Castaño-Vinyals, Marina Pollán, Dora Romaguera, Manolis Kogevinas

**Affiliations:** 1Barcelona Institute for Global Health (ISGlobal), 08003 Barcelona, Spain; ana.espinosa@isglobal.org (A.E.); kurt.straif@isglobal.org (K.S.); gemma.castano@isglobal.org (G.C.-V.); dora.romaguera@isglobal.org (D.R.); manolis.kogevinas@isglobal.org (M.K.); 2Department of Experimental and Health Sciences, Universitat Pompeu Fabra (UPF), 08003 Barcelona, Spain; 3Hospital del Mar Medical Research Institute (IMIM), 08003 Barcelona, Spain; 4Consortium for Biomedical Research in Epidemiology and Public Health (CIBERESP), Institute of Health Carlos III, 28029 Madrid, Spain; bperez@isciii.es (B.P.-G.); ines.gomez@unican.es (I.G.-A.); rocioolmedo@ugr.es (R.O.-R.); alguacil@dbasp.uhu.es (J.A.); fernandeztguillermo@uniovi.es (G.F.-T.); dcasabonne@iconcologia.net (D.C.); nuria.aragones@salud.madrid.org (N.A.); mpollan@isciii.es (M.P.); 5National Centre for Epidemiology, Carlos III Institute of Health, 28029 Madrid, Spain; 6Department of Epidemiology, Centre of Public Health, Medical University of Vienna, 1090 Vienna, Austria; kyriaki.papantoniou@meduniwien.ac.at; 7Facultad de Medicina, Universidad de Cantabria—IDIVAL, 39011 Santander, Spain; 8Cáncer y Salud Pública, FISABIO, 46020 Valencia, Spain; molina_anabar@gva.es; 9Department of Preventive Medicine and Public Health, Universidad de Granada, 18016 Granada, Spain; 10Instituto de Investigación Biosanitaria de Granada (ibs.GRANADA), Hospitales Universitarios de Granada/Universidad de Granada, 18014 Granada, Spain; 11Centro de Investigación en Recursos Naturales, Salud y Medio Ambiente (RENSMA), Universidad de Huelva, Campus Universitario de El Carmen, 21071 Huelva, Spain; 12Instituto de Investigación Sanitaria del Principado de Asturias (ISPA) and IUOPA, Universidad de Oviedo, 33006 Oviedo, Spain; 13Cancer Epidemiology Research Programme IDIBELL, Institut Català d’Oncologia, 08908 L’Hospitalet de Llobregat, Spain; 14Health Research Institute of the Balearic Islands (IdISBa), 07120 Palma de Mallorca, Spain; 15CIBER Fisiopatología de la Obesidad y Nutrición (CIBEROBN), 28029 Madrid, Spain

**Keywords:** prostate cancer, prolonged nighttime fasting, early time-restricted feeding, circadian rhythms, breakfast, chrononutrition

## Abstract

Nighttime fasting has been inconclusively associated with a reduced risk of cancer. The purpose of this study was to investigate this association in relation to prostate cancer risk. We examined data from 607 prostate cancer cases and 848 population controls who had never worked in night shift work from the Spanish multicase-control (MCC) study, 2008–2013. Through an interview, we collected circadian information on meal timing at mid-age. We estimated odds ratios (OR) and 95% confidence intervals (CI) with unconditional logistic regression. After controlling for time of breakfast, fasting for more than 11 h overnight (the median duration among controls) was associated with a reduced risk of prostate cancer compared to those fasting for 11 h or less (OR = 0.77, 95% 0.54–1.07). Combining a long nighttime fasting and an early breakfast was associated with a lower risk of prostate cancer compared to a short nighttime fasting and a late breakfast (OR = 0.54, 95% CI 0.27–1.04). This study suggests that a prolonged nighttime fasting duration and an early breakfast may be associated with a lower risk of prostate cancer. Findings should be interpreted cautiously and add to growing evidence on the importance of chrononutrition in relation to cancer risk.

## 1. Introduction

Prostate cancer is the most frequently diagnosed cancer in men and ranks as the third cause of cancer mortality in Spain [1]. Several non-modifiable risk factors for prostate cancer have been identified, including age, ethnicity and a family history of prostate cancer [2]. In 2019, following a first evaluation in 2007 of shift work that involves circadian disruption, the International Agency for Research on Cancer (IARC) classified night shift work as probably carcinogenic to humans, supported by limited evidence in humans for prostate, breast and colon cancer [3,4]. Circadian rhythms allow the adaptation to daily environmental changes and regulate multiple physiological activities in the organism following cycles of 24 h [5]. In the IARC report, several mechanistic hypotheses were proposed to explain the evaluated association, including chronic inflammation, hormonal alterations, cell proliferation and immunosuppression [3]. In the general population, circadian disruption as a result of exposure to artificial light-at-night (ALAN) [6,7], sleep duration [8,9,10] and mutations in clock genes [11], has also been associated with prostate cancer.

Although the light captured by the retina is the main synchronizer of the central circadian system, the feeding–fasting cycle also plays a major role in the regulation of peripheral clocks. Late-night eating has been associated with an increased prostate cancer risk in a study in France [12]. Similarly, previous results from the multicase-control (MCC) study in Spain found that having a long-time interval between supper and sleep was associated with a statistically significant reduced risk of prostate cancer [13]. This protective association was more pronounced in individuals with a morning chronotype, a human attribute believed to have a genetic basis that reflects a personal preference for timing of activity [13]. The morning chronotype was defined as having a mid-sleep time before 3:35 AM, based on the distribution among controls [13]. Results from the same study showed that having an early supper was associated with a reduced risk of prostate cancer, but results were not statistically significant [13]. Another study in adults reported an association between nighttime snacking and increased body fatness [14], the latter classified as a probable risk factor for prostate cancer [15,16].

Various fasting regimens have been associated with weight loss and have been recently popularized [17]. Intermittent fasting is a form of fasting that consists of restricting the feeding window to 8 h, therefore prolonging the nighttime fasting duration [17]. Prolonged nighttime fasting has also been associated with a reduction in waist circumference, blood glucose (HbA1c) levels and systemic inflammation, which may influence cancer risk [18,19]. Results from the prospective Women’s Healthy Eating and Living study suggested that fasting for 13 h or more reduced the risk of breast cancer recurrence compared to a shorter fasting period [20]. However, data from the NutriNet-Santé prospective cohort study showed that duration of fasting overnight was not significantly associated with the risk of developing prostate cancer [12]. Although nighttime fasting has been associated with a lower risk of cancer outcomes, skipping breakfast has been linked with an increased risk of low-grade inflammation [21].

Food behaviors, including time of supper and interval between supper and sleep, were previously examined in the MCC study [13]. However, the associations with nighttime fasting duration and time of breakfast were not evaluated. Given the current surge in fasting as a new food behavior, in this study we investigate whether prolonged nighttime fasting is associated with a reduced prostate cancer risk Additionally, we consider whether the time window of this period of fasting and the time of breakfast play an important role in this association.

## 2. Methods

### 2.1. Study Population

The MCC study (http://www.mccspain.org, accessed on 6 April 2021) is a multicase-control study conducted in 12 provinces of Spain between 2008 and 2013 [22]. Included in this analysis were individuals between the ages of 20 and 85 with histological confirmation of prostate cancer. As soon as possible after diagnosis, cancer cases were frequency matched based on age, sex and area with population controls. In each of the recruitment areas, administrative records from primary healthcare centers were reviewed to randomly select controls residing there for a minimum period of 6 months. For this study, participants who did not respond to the follow-up circadian interview (*N* = 522) were excluded (Figure 1). To avoid confounding, subjects ever doing night shift work were excluded (*N* = 535), with night shift work defined as working partly or entirely between midnight and 6 AM for 3 nights or more per month [13]. Participants with missing data on nighttime fasting (*N* = 93) were also excluded. In this analysis, 607 prostate cancer cases and 848 population controls from 7 different regions (Madrid, Barcelona, Asturias, Huelva, Cantabria, Valencia and Granada) were selected. Each of the Ethics committees of the included centers reviewed and approved the study protocol. All subjects signed a written informed consent.

### 2.2. Data Collection

A questionnaire was answered by participants in a face-to-face interview with a trained interviewer [22]. We requested information on age, sociodemographic factors, family history of prostate cancer, smoking, sleep duration, sleep problems and height and weight one year before the interview, from which the body mass index (BMI) was calculated as kg/m^2^. Exposure to indoor ALAN was assessed in the questionnaire through a Likert scale ranging from total darkness, almost dark and dim light to quite illuminated [6]. Outdoor ALAN exposure was also assessed but only for the cities of Barcelona and Madrid, where satellite images were available [6]. Light exposure was modeled and attributed to participants’ geocoded residences through Geographic Information System [6].

Following the interview, information on daily energy intake was gathered through a semi-quantitative food frequency questionnaire answered at home and estimated using the Centro de Enseñanza Superior de Nutrición y Dietética (CESNID) food composition table [23]. Ten percent of the participants did not respond to the food frequency questionnaire. A World Cancer Research Fund/American Institute of Cancer Research (WCRF/AICR) score was constructed [24]. The WCRF/AICR score ranged from 0 to 6 and included information on body fatness, physical activity, foods and drinks that promote weight gain, plant foods, animal foods and alcoholic drinks. This variable was further categorized into sex-specific tertiles based on the distribution in the control group [24].

Circadian data, including information on meal frequencies, duration and timings, were assessed through a telephone interview conducted 6 months to 5 years after the initial interview (median time 3 years) [13]. In this paper, we refer to these variables as dietary circadian variables. Participants reported their dietary circadian variables on weekdays and weekends at 40 years of age and during the year prior to the initial interview. Participants younger than 40 years of age were only asked for the information on the year prior to recruitment. Participants were also asked questions on sleep and timing of physical activity and answered the Munich Chronotype Questionnaire [13]. Chronotype was estimated as the standard mid-sleep time on free days, Mid-sleep time on free days (MSF) = [sleep onset on a free day + (sleep duration on free day/2)], and mid-sleep time on free days corrected for oversleeping on free days (MSFcorr) = MSF − [sleep duration on a free day − (sleep duration on a working day/2)]. Further details have been described elsewhere [13].

### 2.3. Exposure and Outcome Assessment

Nighttime fasting duration was defined as the period elapsed between the last eating episode before going to sleep and the first episode the following day. In the previous paper from MCC, we investigated time of supper [13], but now time of last intake also considered after supper snacks. For participants reporting not having breakfast, lunch was considered as breakfast, understood as the wider concept of breaking the nightly fast. The cut-off point for nighttime fasting distribution was set at 11 h (the median distribution among controls), defining a categorical variable with two levels: 11 h or less of fast, the reference category, versus more than 11 h of fast. We based the main analysis on patterns of weekdays and at 40 years of age, because weekdays are more representative of daily lifestyle habits, and this also helped avoid having missing data on weekend patterns. We used data at 40 years of age to avoid potential reverse causation since it has been reported that cancer can cause anorexia (loss of appetite), thus affecting eating patterns [25].

Clinical information on prostate cancer aggressiveness determined by the Gleason score was recorded from medical records. We did not use the new Gleason grading system because of small numbers in some groups [26]. We classified cases into two groups: low-grade aggressive prostate cancer (Gleason score = 6 or 3 + 4) and high-grade aggressive prostate cancer (Gleason score 4 + 3 or higher), as it has been previously reported in the literature [27].

### 2.4. Statistical Analyses

We compared the distribution of characteristics in prostate cancer cases and controls and also prostate cancer risk factors according to nighttime fasting duration in controls. To evaluate significant differences, chi-square tests and *t*-tests were applied to calculate the *p*-value in categorical and continuous variables, respectively.

We assessed the correlation between nighttime fasting and the other dietary circadian variables (time of supper, interval between supper and sleep and time of breakfast) using Spearman’s correlation coefficient. We examined the linearity of the association between nighttime fasting duration (continuous variable, in hours) and prostate cancer using generalized additive models (GAM). We applied ANOVA, including the smoothing terms versus the linear term model, to test this association’s linearity.

Odds ratios (OR) and 95% confidence intervals (CI) were estimated with unconditional logistic regression using categorical exposure data. The crude model was adjusted for age (continuous, years), center (Madrid, Barcelona, Asturias, Huelva, Cantabria, Valencia, Granada) and for educational level (less than primary school, primary school, secondary school, university). We included educational level in the crude model because population controls had a higher level of education compared to cases [22]. Moreover, in our study population, education was significantly associated both with the exposure and the outcome (see Table 1 and Table 2).

Since there is limited prior evidence of determinants for both the exposure and the outcome, we followed a mixed criterion for the selection of further potential confounders. First, on the basis of the limited prior knowledge, we built a directed acyclic graph using Daggity (Appendix A) [28]. The variables considered were family history of prostate cancer (yes, no), adherence to a healthy lifestyle through the WCRF/AICR score (high, medium, low), indoor ALAN (total darkness, almost dark, dim light, quite illuminated), outdoor ALAN (continuous, cd/m^2^), sleep problems (yes, no) and duration (7 h or less, more than 7 h), number of daily eating episodes (three or less, more than three), time of the last intake (before 9 PM, 9:00 to <10 PM, 10 PM, or later) and the interval between the time of supper and sleep (1 h or less, from >1 to 2 h, more than 2 h). “Time of breakfast” was categorized in two levels (8:30 AM or before, after 8:30 or skip breakfast). Skipping breakfast has been associated with negative metabolic outcomes, but in our study only 3 cases reported never having breakfast. We also considered timing of physical activity (inactive, 8 AM–10 AM, 10 AM–12 PM, 12 PM–7 PM, 7 PM–11 PM, any other pattern), since it has been recently reported that morning exercise may be associated with a lower risk of prostate cancer compared to exercising later in the day [29]. Additionally, we considered diabetes (yes, no) because its treatment can include changes in dietary habits and it has been inconsistently suggested as a protective factor for prostate cancer [30,31,32]. Finally, we included BMI (continuous, kg/m^2^), smoking (never, ex-smoker, current smoker) and chronotype (morning, intermediate, evening).

In the adjusted model we included variables strongly associated with both the exposure and outcome in our study population (see Table 1 and Table 2) and variables changing the association between nighttime fasting and prostate cancer risk by more than 10% (Appendix A). These variables were diabetes and indoor ALAN for the first criterion and breakfast time as the second criterion. Less than 1% of data was missing in the selected covariates. Multicollinearity was assessed in the adjusted model through the variance inflation factor to control for redundancy.

We explored whether there was evidence of effect modification of the association of interest by chronotype, healthy lifestyle (assessed with the WCRF/AICR score) and time of breakfast, including an interaction term in each of the models and examining results from a likelihood ratio test. We also investigated the combined association of nighttime fasting duration and time of breakfast with prostate cancer risk. The association between nighttime fasting and prostate cancer aggressiveness was examined with a multinomial logistic regression model. The statistical package R-4.0.0 (The R Project for Statistical Computing, Vienna, Austria) was used for these analyses.

### 2.5. Sensitivity Analyses

We explored the role of nighttime fasting on days off and working days by combining nighttime fasting data from weekdays and weekends (at 40 years of age). We used a weighted mean giving a weight of 5/7 to weekdays and of 2/7 to weekends. We also explored eliminating point outliers of the exposure distribution to exclude the most extreme patterns. We considered as potential point outliers any observation below the Q1 − (1.5*IQR) and above Q3 + (1.5*IQR). Nine (0.6%) and 35 observations (2.4%) were identified and excluded, respectively. To test the robustness of our results, exposure the previous year to the circadian interview was also considered, despite the potential bias due to reverse causation. This model was adjusted for time of breakfast corresponding to the year before the circadian interview and we assumed that exposure to indoor ALAN did not change during these years.

The time elapsed between the inclusion in the study and the exposure assessment from the circadian interview was long for some participants. Therefore, we did a sensitivity analysis including only those participants that answered the interview 3 years or less after the baseline questionnaire.

## 3. Results

### 3.1. Study Population

Characteristics of the included prostate cancer cases (*N* = 607) and matched controls (*N* = 848) are shown in Table 1. Educational level (*p*-value < 0.001), family history of prostate cancer (*p*-value < 0.001) and poor adherence to the WCRF/AICR recommendations (*p*-value = 0.004) were significantly associated with prostate cancer. Having a shorter time interval elapsed between supper and sleep (*p*-value = 0.034), not having breakfast consistently (*p*-value = 0.001) and being exposed to indoor artificial light at night (*p*-value = 0.001) were significantly associated with prostate cancer. Cases were less likely to have diabetes (*p*-value = 0.003).

### 3.2. Nighttime Fasting and Prostate Cancer Risk

The median nighttime fasting duration in the control group was 11 h (interquartile range 10–12). The distribution among controls of prostate cancer risk factors according to categories of nighttime fasting duration is presented in Table 2. Highly educated subjects tended to fast fewer hours overnight (*p*-value < 0.001). Participants with diabetes and less exposed to indoor ALAN tended to have more extended nighttime fasting periods (*p*-value = 0.052 and 0.005, respectively). Participants who never had breakfast or had the first intake after 8:30 AM and the last intake before 9 PM tended to fast for more hours (*p*-value < 0.001 for both variables).

A low positive correlation was observed between nighttime fasting duration and time elapsed between supper and sleep (Appendix A, with a Spearman’s correlation coefficient of 0.12). Time of supper and nighttime fasting duration showed a low negative correlation (Appendix A, with a Spearman’s correlation coefficient of −0.36). High correlation was found between nighttime fasting duration and time of breakfast (Appendix A, Spearman’s correlation coefficient of 0.83).

The GAM showed a slight reduction of prostate cancer risk with more extended nighttime fasting (Appendix A). When adjusting this model for time of breakfast, the reduction in the prostate cancer risk associated with fasting was more evident (Appendix A). In the crude logistic regression model, fasting for more than 11 h overnight was associated with a slightly non-significant reduced risk of prostate cancer compared to those fasting for 11 h or less (OR = 0.92, 95% CI 0.73–1.16, Table 3). After adjusting for confounders, the model showed that a more extended nightly fast was linked to a more potent reduction of prostate cancer risk (adjusted model, OR = 0.77, 95% CI 0.54–1.07, Table 3). In this model, having breakfast after 8:30 AM was associated with a non-significant increased risk of prostate cancer (OR = 1.30, 95% CI 0.92–1.85) compared to having breakfast at 8:30 AM or before. We did not find important collinearity in the adjusted model between the two diet time-related variables.

We also tried categorizing the exposure into three categories based on the distribution of this variable in the control group: 10 h or less (reference category), more than 10 h to 12 h, and more than 12 h (Appendix A). In this analysis, fasting for more than 10 h to 12 h was associated with a slightly higher prostate cancer reduced risk than the most extended fasting category. This difference was dissipated when adjusting for time of breakfast, suggesting that this difference could be explained by the adverse effects of postponing breakfast.

### 3.3. Prostate Cancer Risk Stratified by Time of First Intake

The prostate cancer risk reduction with a prolonged nightly fast was moderately stronger among those individuals having their breakfast at 8:30 or before (adjusted model, OR = 0.60, 95% CI 0.33–1.04) compared to those having it later on the day or skipping it (OR = 0.90, 95% CI 0.58–1.39, Table 4). Despite this pattern, confidence intervals overlapped and the interaction in the adjusted model was not significant (*p*-value = 0.26, 1 degree of freedom).

### 3.4. Association Combining Nighttime Fasting Duration and Time of Breakfast with Prostate Cancer Risk

When combining both nighttime fasting duration and time of breakfast we observed that the combination associated with a lower risk of prostate cancer was having a long nighttime fasting and an early breakfast compared to a short nighttime fasting and a late breakfast (OR = 0.54, 95% CI 0.27–1.04, Table 5). This association was maintained even after adjusting for time of last intake and interval between supper and sleep.

### 3.5. Prostate Cancer Risk Stratified by Chronotype

The association of prolonged fasting period overnight and reduced prostate cancer risk was slightly more pronounced in morning chronotype individuals (adjusted model, OR = 0.70, 95% CI 0.47–1.04) compared to intermediate and particularly evening ones (OR= 0.80, 95% CI 0.50–1.27; OR = 0.99, 95% CI 0.51–1.92, Table 6). However, the likelihood ratio test for interaction in the adjusted model showed a *p*-value = 0.60 with 2 degrees of freedom. On the other hand, we observed no effect modification by adherence to the WCRF/AICR score (Appendix A).

### 3.6. Relative Risk among Cancer Subtypes

We evaluated the association between fasting overnight and aggressiveness of prostate cancer (Gleason score) in a multinomial logistic regression model. The association of prolonged fasting overnight and reduced prostate cancer risk was similar in cases diagnosed with a more aggressive prostate cancer subtype (Gleason score of 4 + 3 or higher) and in those with a Gleason score of 6 or 3 + 4 (adjusted model, OR = 0.78, 95% CI 0.53–1.14 and OR = 0.71, 95% CI 0.42–1.19, Table 7). The differences among cancer subtypes were non-significant (*p*-value = 0.97).

### 3.7. Sensitivity Analyses

Further adjustment of the model with other potential confounders did not change the estimates (Appendix A). Combining data from weekdays and weekends did not result in different estimates than on weekdays only (data not shown). Removing the outliers of the exposure variable distribution also had a minimal effect on estimates (adjusted model, OR = 0.76, 95% CI 0.54–1.06, Appendix A). The GAM in this reduced dataset showed the same pattern (see Appendix A).

The median nighttime fasting duration reported one year before the inclusion in the study in the control group was 11.50 h (interquartile range 10.50–12.50). Models built with data on nighttime fasting during weekdays reported one year before diagnosis or before inclusion in the study for the controls changed the direction of the association (adjusted model OR = 1.13, 95% CI 0.88–1.46, Appendix A). Analyzing the cancer subtypes, we observed that the relative risk ratio for aggressive prostate cancer cases (adjusted RRR = 1.23, 95% CI 0.82–1.84, Appendix A) was slightly higher than for less aggressive cases (adjusted RRR = 1.10, 95% CI 0.83–1.46, Appendix A).

In a model that excluded participants answering the circadian interview more than 3 years later from the baseline interview strengthened the findings (participants interviewed within 3 years: OR = 0.68, 95% CI 0.41–1.13; all participants: OR = 0.77, 95% CI 0.54–1.07).

## 4. Discussion

This is one of the first studies to examine the association between nighttime fasting duration and cancer risk. These findings suggest a possible lower risk of prostate cancer with longer overnight fasting time, especially when feeding is synchronized with the appropriate circadian timing by having an early breakfast.

From an evolutionary perspective, the benefits of a prolonged period of fast overnight are plausible. Life on earth has evolved to adapt and anticipate daily environmental oscillations resulting from the 24 h rotation of our planet. Natural selection drove the development of circadian rhythms enabling the daily optimization of energy acquisition, including sunlight (for photosynthetic organisms) or food [33]. The circadian clock permitted organisms to be prepared and aroused to access food when available and store supplies and undergo repairment processes during fasting hours, improving metabolic fitness and survival [33]. The extensive discussion on the benefits of nighttime fasting in mass media (e.g., Twitter) that are frequently based on this evolutionary perspective is not, however, backed by reliable population data.

Prolonged periods of fast overnight have been linked to a significant improvement in glycemic control and inflammation biomarkers, potentially explaining a lower risk of prostate cancer [18,19]. Our results are consistent with those of the prospective WHEL study, showing that fasting for less than 13 h was associated with a 36% increased hazard of breast cancer recurrence than a more extended fasting period [20]. Similarly to what we found, the prospective study from the NutriNet-Santé cohort showed that each hour increase in time of first intake was associated with a 17% increase in the hazard of developing PCa (HR = 1.17, 95% CI 0.96–1.43) [12]. However, contrary to what we found, even adjusting for this confounding effect of time of breakfast, nighttime fasting duration was associated with a slight increase in the hazard of developing prostate cancer in this study population (HR= 1.12, 95% CI 0.93–1.34) [12]. Differences between findings of epidemiological studies might be explained because of varying methods, sample sizes and time intervals examined.

It has been previously reported that time of supper [12,13] and time elapsed between supper and sleep [13] are associated with breast and prostate cancer. To disentangle these effects from the effects of nighttime fasting, we inspected their correlation and we explored adjustment of the crude model for these variables. We found low correlation between nighttime fasting duration and time of supper and with time interval between supper and sleep. Similarly, when we explored adjusting our basic model for time of last intake and for time interval between supper and sleep, the estimates did not significantly change. These results show that the potentially protective association between a prolonged nightly fasting period and reduced prostate cancer risk might be complementary to the protective association of having a larger time interval between supper and sleep.

Fasting regimens that promote long periods of fasting overnight, even by skipping breakfast, have become increasingly popular. Our results suggest that the time of breakfast is confounding the association between nighttime fasting and prostate cancer risk and that elongating the nightly period of fast by skipping breakfast might be counterproductive. Nighttime fasting is most beneficial when the eating window starts early in the morning, a form of intermittent fasting known as early time-restricted feeding [34]. Time-of-day variations in the benefits of nightly fasting can be explained by fluctuations of the circadian rhythms, which can also vary across chronotypes. The optimal time for food intake might be in the early hours of the day when diet-induced thermogenesis, glucose tolerance, insulin sensitivity, pancreatic beta-cell responsiveness and oxidation of fatty acids are higher [35].

Our study shows that in morning chronotype individuals, the association between a prolonged period overnight and a prostate cancer risk reduction is slightly larger than other chronotypes. This association was independent of the time of the first intake. On the contrary, in evening chronotype individuals, long nighttime fasting was not associated with lower prostate cancer risk. This may be explained by a higher frequency of chronic sleep deprivation and social jet-lag among late chronotypes, especially when working on early morning work schedules [36].

Reverse causation might explain the differences observed between the analyses of habits at 40 years of age and those shortly before disease or inclusion in the study. A possible explanation may be that eating and sleep habits one year before diagnosis might be reflecting a cancer-related loss of appetite or other early disease-related changes in lifestyle [25].

The strengths of this research include the novelty of the research question, the large sample size, the adjustment for a wide variety of well-measured confounders and the validated questionnaires to assess diet and the individual chronotype. As limitations, one of the primary concerns is that this assessment is subject to recall bias [13]. We tried to minimize this bias by requesting data on the timing of diet at mid-life. Additionally, in a sensitivity analysis, we explored excluding those participants with a longer time elapsed between inclusion in the study and exposure assessment in the circadian interview and, therefore, more prone to recall bias. After excluding these participants, estimates for our association of interest were strengthened. Meal timings and diet quality were assessed in one time point and these are behaviors prone to fluctuations. Future studies should include multiple time assessments to increase the validity of the results. A further inherent limitation of studies on prostate cancer is the lack of knowledge of this cancer’s non-genetic risk factors that could result in uncontrolled confounding. Finally, circadian patterns, including sleep timing and duration, light exposure and meal timings, are all interconnected. To improve public health recommendations, a future approach could be to analyze these behaviors in an integrated manner.

## 5. Conclusions

To conclude, this is one of the first studies with epidemiological data examining the association between nighttime fasting duration and prostate cancer risk and to consider the time window of this period of fasting. Complementing the previous results from the MCC study that showed a beneficial association between an early supper and a long supper–sleep time interval, our results suggest that lengthening the fasting period could be associated with a lower risk of this cancer, especially when having an early breakfast. This research highlights the importance of considering a circadian perspective, such as the time window of nighttime fasting and combined effects of meal timing aspects, in studies evaluating dietary and sleep determinants of cancer.

## Figures and Tables

**Figure 1 nutrients-13-02662-f001:**
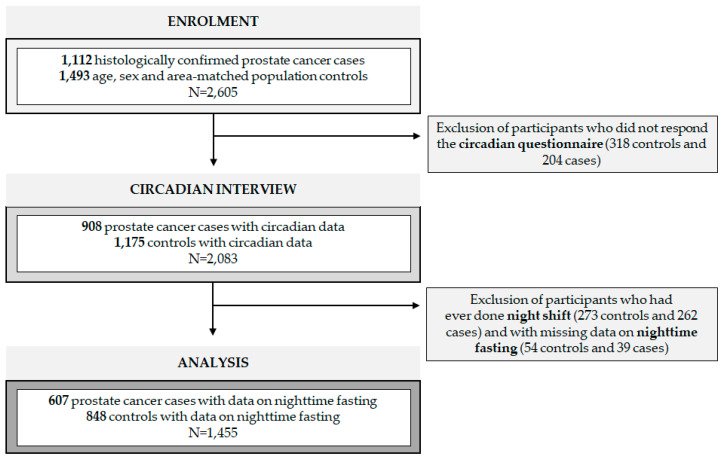
Flow chart of study population. N = sample size.

**Table 1 nutrients-13-02662-t001:** Basic characteristics of study population.

	Controls (*N* = 848)	Cases (*N* = 607)	*p*-Value
	Mean (SD) or *N* (%)	Mean (SD) or *N* (%)	
**Age**	66.0 (8.4)	65.6 (7.0)	0.372
**BMI**	27.5 (3.6)	27.5 (3.6)	0.967
Normal weight (<25 kg/m^2^)	225 (26.5)	153 (25.2)	0.850
Overweight (≥25 to <30 kg/m^2^)	439 (51.8)	320 (52.7)	
Obese (≥30 kg/m^2^)	184 (21.7)	134 (22.1)	
**Educational level**			
Less than primary school	115 (13.6)	104 (17.1)	<0.001
Primary school	247 (29.1)	244 (40.2)	
Secondary school	264 (31.1)	142 (23.4)	
University	222 (26.2)	117 (19.3)	
**Family history of prostate cancer**			
No	793 (93.5)	506 (83.4)	<0.001
Yes	55 (6.5)	101 (16.6)	
**Smoking**			
Never	242 (28.5)	171 (28.2)	0.571
Ex-smoker	434 (51.2)	325 (53.5)	
Current smoker	172 (20.3)	111 (18.3)	
**Chronotype**			
Morning	419 (50.5)	306 (50.5)	0.974
Intermediate	303 (36.6)	224 (37.0)	
Evening	107 (12.9)	76 (12.5)	
Unknown	19	1	
**WCRF/AICR score**			
Low adherence ^a^	307 (40.1)	203 (37.3)	0.004
Medium adherence ^b^	262 (34.2)	232 (42.6)	
High adherence ^c^	196 (25.6)	109 (20.0)	
Unknown	83	63	
**Diabetes**			
No	671 (79.3)	518 (85.6)	0.003
Yes	175 (20.7)	87 (14.4)	
Unknown	2	2	
**Indoor ALAN exposure**			
Total darkness	147 (17.4)	87 (14.4)	0.001
Almost dark	348 (41.2)	211 (34.9)	
Dim light	261 (30.9)	204 (33.7)	
Quite illuminated	88 (10.4)	103 (17.0)	
Unknown	4	2	
**Breakfast**			
No	17 (2.0)	3 (0.5)	0.001
Only weekends	3 (0.4)	12 (2.0)	
Only weekdays	15 (1.8)	16 (2.6)	
Always	809 (95.9)	575 (94.9)	
Unknown	4	1	
**Time of breakfast**			
8:30 AM or before	466 (55.0)	314 (51.7)	0.227
After 8:30 AM or skip breakfast	382 (45.0)	293 (48.3)	
**Time of last intake**			
10 PM or later	294 (34.7)	227 (37.4)	0.350
9:00 to <10 PM	439 (51.8)	311 (51.2)	
Before 9 PM	115 (13.6)	69 (11.4)	
**Supper/sleep interval**			
1 h or less	186 (22.2)	169 (28.1)	0.034
From >1 to ≤2 h	327 (39.0)	223 (37.0)	
More than 2 h	326 (38.9)	210 (34.9)	
Unknown	9	5	

ALAN = artificial light at night; BMI = body mass index; *N* = sample size; NA = not applicable; SD = standard deviation; WCRF/AICR = World Cancer Research Fund / American Institute for Cancer Research. ^a^ Men (0.25–3); Women (0.5–3.5). ^b^ Men (3.25–4); Women (3.75–4.25). ^c^ Men (4.25–6); Women (4.5–6).

**Table 2 nutrients-13-02662-t002:** Distribution of prostate cancer risk factors according to nighttime fasting duration in controls.

	≤11 h of Fast (*N* = 474)	>11 h of Fast (*N* = 374)	*p*-Value
	Mean (SD) or *N* (%)	Mean (SD) or *N* (%)	
**Age**	65.5 (8.5)	66.5 (8.2)	0.070
**BMI**	27.4 (3.5)	27.6 (3.8)	0.384
Normal weight (<25 kg/m^2^)	121 (25.5)	104 (27.8)	0.198
Overweight (≥25 to <30 kg/m^2^)	258 (54.4)	181 (48.4)	
Obese (≥30 kg/m^2^)	95 (20.0)	89 (23.8)	
**Educational level**			
Less than primary school	50 (10.5)	65 (17.4)	<0.001
Primary school	115 (24.3)	132 (35.3)	
Secondary school	161 (34.0)	103 (27.5)	
University	148 (31.2)	74 (19.8)	
**Family history of prostate cancer**			
No	445 (93.9)	348 (93.0)	0.727
Yes	29 (6.1)	26 (7.0)	
**Smoking**			
Never	140 (29.5)	102 (27.3)	0.191
Ex-smoker	230 (48.5)	204 (54.5)	
Current smoker	104 (21.9)	68 (18.2)	
**Chronotype**			
Morning	219 (47.1)	200 (54.9)	0.080
Intermediate	181 (38.9)	122 (33.5)	
Evening	65 (14.0)	42 (11.5)	
Unknown	9	10	
**WCRF/AICR score**			
Low adherence ^a^	164 (38.2)	143 (42.6)	0.432
Medium adherence ^b^	154 (35.9)	108 (32.1)	
High adherence ^c^	111 (25.9)	85 (25.3)	
Unknown	45	38	
**Diabetes**			
No	387 (81.8)	284 (76.1)	0.052
Yes	86 (18.2)	89 (23.9)	
Unknown	1	1	
**Indoor ALAN exposure**			
Total darkness	63 (13.3)	84 (22.6)	0.005
Almost dark	203 (43.0	145 (39.0)	
Dim light	151 (32.0)	110 (29.6)	
Quite illuminated	55 (11.7)	33 (8.9)	
Unknown	2	2	
**Breakfast**			
No	1 (0.2)	16 (4.3)	<0.001
Only weekends	NA	3 (0.8)	
Only weekdays	8 (1.7)	7 (1.9)	
Always	465 (98.1)	344 (93.0)	
Unknown	NA	4	
**Time of breakfast**			
8:30 AM or before	417 (88.0)	49 (13.1)	<0.001
After 8:30 AM or skip breakfast	57 (12.0)	325 (86.9)	
**Time of last intake**			
10 PM or later	215 (45.4)	79 (21.1)	<0.001
9:00 to <10 PM	233 (49.2)	206 (55.1)	
Before 9 PM	26 (5.5)	89 (23.8)	
**Supper/sleep interval**			
1 h or less	110 (23.6)	76 (20.4)	0.553
From >1 to ≤2 h	178 (38.1)	149 (40.1)	
More than 2 h	179 (38.3)	147 (39.5)	
Unknown	7	2	

ALAN = artificial light at night; BMI = body mass index; *N* = sample size; SD = standard deviation; WCRF/AICR = World Cancer Research Fund / American Institute for Cancer Research. ^a^ Men (0.25–3); Women (0.5–3.5). ^b^ Men (3.25–4); Women (3.75–4.25). ^c^ Men (4.25–6); Women (4.5–6).

**Table 3 nutrients-13-02662-t003:** Association of nighttime fasting and prostate cancer risk.

Nighttime Fasting	Controls *N* (%)	Cases *N* (%)	OR (95% CI) ^a^	OR (95% CI) ^b^
≤11 h	474 (55.9)	342 (56.3)	Ref	Ref
>11 h	374 (44.1)	265 (43.7)	0.92 (0.73–1.16)	0.77 (0.54–1.07)

*N* = sample size; OR = odds ratio; 95% CI = 95% confidence interval. ^a^ Adjusted for age, center and education. ^b^ Adjusted for age, center, education, diabetes (missing for 2 controls and 2 cases), indoor ALAN exposure (missing for 4 controls and 2 cases) and time of breakfast.

**Table 4 nutrients-13-02662-t004:** Association of nighttime fasting and prostate cancer risk stratified by time of breakfast.

Nighttime Fasting	Controls *N* (%)	Cases *N* (%)	OR (95% CI) ^a^	OR (95% CI) ^b^
Breakfast at 8:30 AM or before				
≤11 h	417 (89.5)	293 (93.3)	Ref	Ref
>11 h	49 (10.5)	21 (6.7)	0.66 (0.37–1.13)	0.60 (0.33–1.04)
Breakfast after 8:30 AM or skip breakfast				
≤11 h	57 (14.9)	49 (16.7)	Ref	Ref
>11 h	325 (85.1)	245 (83.3)	0.87 (0.57–1.33)	0.90 (0.58–1.39)

*N* = sample size; OR = odds ratio; 95% CI = 95% confidence interval. ^a^ Adjusted for age, center and education. ^b^ Adjusted for age, center, education, diabetes (missing for 2 controls and 2 cases), indoor ALAN exposure (missing for 4 controls and 2 cases) and chronotype (missing for 19 controls and 1 case).

**Table 5 nutrients-13-02662-t005:** Association of nighttime fasting and time of breakfast with prostate cancer risk.

	Controls *N* (%)	Cases *N* (%)	OR (95% CI) ^a^	OR (95% CI) ^b^
Short nighttime fasting (≤11 h) and late breakfast (>8:30 AM)	56 (6.6)	49 (8.1)	Ref	Ref
Short nighttime fasting (≤11 h) and early breakfast (≤8:30 AM)	418 (49.3)	293 (48.3)	0.88 (0.57–1.36)	0.89 (0.57–1.39)
Long nighttime fasting (>11 h) and late breakfast (>8:30 AM)	325 (38.3)	244 (40.2)	0.87 (0.57–1.33)	0.88 (0.57–1.36)
Long nighttime fasting (>11 h) and early breakfast (≤8:30 AM)	49 (5.8)	21 (3.5)	0.58 (0.30–1.11)	0.54 (0.27–1.04)

*N* = sample size; OR = odds ratio; 95% CI = 95% confidence interval. ^a^ Adjusted for age, center and education. ^b^ Adjusted for age, center, education, diabetes (missing for 2 controls and 2 cases) and indoor ALAN exposure (missing for 4 controls and 2 cases).

**Table 6 nutrients-13-02662-t006:** Association of nighttime fasting and prostate cancer risk stratified by chronotype (missing for 19 controls and for 1 case).

Nighttime Fasting	Controls *N* (%)	Cases *N* (%)	OR (95% CI) ^a^	OR (95% CI) ^b^
Morning chronotype				
≤11 h	219 (52.3)	170 (55.5)	Ref	Ref
>11 h	200 (47.7)	136 (44.5)	0.84 (0.61–1.15)	0.70 (0.47–1.04)
Intermediate chronotype				
≤11 h	181 (59.7)	130 (58.0)	Ref	Ref
>11 h	122 (40.3)	94 (42.0)	0.97 (0.67–1.41)	0.80 (0.50–1.27)
Evening chronotype				
≤11 h	65 (60.7)	41 (53.9)	Ref	Ref
>11 h	42 (39.3)	35 (46.1)	1.17 (0.63–2.18)	0.99 (0.51–1.92)

*N* = sample size; OR = odds ratio; 95% CI = 95% confidence interval. ^a^ Adjusted for age, center and education. ^b^ Adjusted for age, center, education, diabetes (missing for 2 controls and 2 cases), indoor ALAN exposure (missing for 4 controls and 2 cases) and time of first intake.

**Table 7 nutrients-13-02662-t007:** Association of nighttime fasting and prostate cancer relative risk by cancer subtypes (missing for 17 cases).

Nighttime Fasting	Controls *N* (%)	Cases *N* (%)	RRR (95% CI) ^a^	RRR (95% CI) ^b^
Low aggressiveness				
≤11 h	474 (55.9)	246 (56.7)	Ref	Ref
>11 h	374 (44.1)	188 (43.3)	0.91 (0.71–1.18)	0.78 (0.53–1.14)
High aggressiveness				
≤11 h	474 (55.9)	86 (55.1)	Ref	Ref
>11 h	374 (44.1)	70 (44.9)	0.92 (0.63–1.33)	0.71 (0.42–1.19)

*N* = sample size; RRR = Relative risk ratio; 95% CI = 95% confidence interval. ^a^ Adjusted for age, center and education. ^b^ Adjusted for age, center, education, diabetes (missing for 2 controls and 2 cases), indoor ALAN exposure (missing for 4 controls and 2 cases) and time of first intake.

## Data Availability

Data can be provided by contacting the corresponding author and following permission by the participating centres in the MCC Spain study. The computing code can be provided by the corresponding author.

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
