# Peer review of "The Association of Nighttime Fasting Duration and Prostate Cancer Risk: Results from the Multicase-Control (MCC) Study in Spain"

_nutrients, 2021, doi:10.3390/nu13082662_

Round 1
Reviewer 1 Report
Palomar-Cros et al. present an assessment of the effects of nighttime fasting duration on prostate cancer risk in the MCC-Spain cohort. This is an informative cohort that has already been analysed from many different angles. Overall, evaluation of chrononutrition parameters in such cohorts is important. Especially, assessing the impact of night time fasting duration on certain pathologies is relevant in the context of other preclinical time-restricted feeding studies (lab of S. Panda). Finally, the article is well written. However, this reviewer has one major concern that need to be clarified before the article can be published.
Major concern:
The authors need to clearly differentiate their analyses in the submitted manuscript (MS) from reports on the MCC-Spain cohort published before. In particular, a clear distinction from their 2018 report in Int J of Cancer (Ref 13 in the present MS) needs to be worked out early in the article (abstract and introduction). Although the authors mention the earlier paper, it remains obscure how analyses done for the present MS are different from the Int J of Cancer paper where many parameters where already measured! Hence, it needs clarification if and how the cohorts of the two studies overlap (same subjects?) and which analyses have been done in addition to the Int J of Cancer paper. In that respect, the paper has to be rewritten to explain the merit of this new analyses and to prevent double-reporting (example: supper/sleep interval was assessed in both studies).
Minor points:
- Table 1: Time of last intake is not significant although this was one of the major outcomes of their 2018 study. How can this be reconciled?
- Can the authors justify why the adjusted for education and discuss, how education could factor into cancer risk.
- The same goes for diabetes although here it is an inverse relation.
Reviewer 2 Report
Review report:
Manuscript titled: The association of nighttime fasting duration and prostate cancer risk in Spain, authored by Palomar-Cros et al.
The revised manuscript aimed to assess whether the prolonged nighttime fasting is associated with a reduced prostate cancer risk. In my opinion the obtained results are very interesting and address new potential risk factors for prostate cancer risk, but before consideration the manuscript in publication in Nutrients Journal, the text requires some improvements.
My decision: major revision
Detailed comments and suggestions:
- the tile of manuscript should be changed. I recommend: “The association of nighttime duration and prostate cancer (PC) risk – the results from multicase-control study from Spain”
- The abstract did not include the purpose of the study
- Introduction: the 3rd sentence needs a relevant reference; line 7, in the sentence: “the body” should be replaced by “organism”; please define, what do you mean by “morning chronotype”? What are the potential mechanisms of diurnal rhythms disruption and prostate tumorigenesis? What are the probable factors responsible for that process? (please add the relevant information to this section)
- Methods: the flow-chart of study population should be moved from supplementary material to this section!
- Results: The results of the study are described correctly, but please pay an attention that explanation of abbreviations in the description under the tables should be adapted to their content. Did you record information about food consumption from the study participants? BMI should be also expressed categorically (normal weight subjects, overweighted and obese subjects)
Could the association of nighttime fasting and prostate cancer be modified by a recognized PC factors , for example, high intake of alcohol, processed red meat products or whole milk products (or not). This issue should be considered also in Discussion.
- Please define the main study limitations.
Round 2
Reviewer 1 Report
The authors sufficiently addressed my concerns and the implemented changes and rewordings warrant publication.
Reviewer 2 Report
The Authors addressed all my comment and suggestions. I accept the manuscript at present form for publication.